# The Efficiency of Bimodal Silica as a Carbon Dioxide Adsorbent for Natural Gas Treatment

**Fabíola Correia de Carvalho [1,2,]*** , **Paula Fabiane do Nascimento [3]** ,
**Márcio Rodrigo Oliveira de Souza [1]** and **Antonio Souza Araujo [1]**

[1] Postgraduate Program in Petroleum Science and Engineering, Federal University of Rio Grande do Norte, Natal 59078-970, RN, Brazil; marciorodrigo2006@hotmail.com (M.R.O.d.S.); araujo.ufrn@gmail.com (A.S.A.)
[2] SENAI Institute of Innovation in Renewable Energy, Sustainability Laboratory, Natal 59064-164, RN, Brazil
[3] Postgraduate Program in Chemical Engineering, Federal University of Rio Grande do Norte, Natal 59078-970, RN, Brazil; paula_fabiane@hotmail.com
* Correspondence: fabiola_correia@hotmail.com; Tel./Fax: +55-849-992-979-23

**Abstract:** Natural gas (NG) production in Brazil has shown a significant increase in recent years. Oil and natural gas exploration and refining activities indicate circa 86% carbon dioxide content in NG, representing a serious problem for environmental issues related to greenhouse gas emissions and increases in global warming. New technologies using $CO_2$ capture materials have been shown to be more efficient than conventional processes. In this work, a bimodal meso–macroporous silica adsorbent for $CO_2$ adsorption in NG was synthesized and evaluated as a promising material for use in natural gas treatment systems, as silica has specific textural properties that facilitate the capture and storage of this gas. The adsorbent was obtained from silica via the hydrothermal method with n-dodecane emulsion and characterized by X-ray diffraction, scanning electron microscopy, infrared spectroscopy, and the BET specific surface area. Adsorption capacity tests were performed for $CO_2$, methane, and their mixtures by the gravimetric method, demonstrating that the adsorbent was selective for $CO_2$ and obtained a good adsorption capacity. The experimental values obtained were compared and adjusted to the models of Langmuir and Freundlich. Thus, the bimodal silica adsorbent developed in this research proved to be excellent for $CO_2$ adsorption and is a promising material for the treatment of NG.

**Keywords:** carbon dioxide adsorption; methane; natural gas; silica; hybrid materials

## 1. Introduction

Natural gas (NG) production in Brazil has shown a significant increase in recent years [1] and comprises a mixture of gaseous hydrocarbons, mainly methane, followed by ethane, propane, butane, and heavier hydrocarbons. In addition to these hydrocarbons, other compounds may be present, such as water, nitrogen, hydrogen sulfide ($H_2S$), carbon dioxide ($CO_2$), other sulfur compounds, and impurities, with $H_2S$ and $CO_2$ being the most common. These problems, in addition to causing corrosion and the leakage of pipes, decrease the quality of the fuel, necessitating the effective treatment of NG to remove these compounds [2].

Oil and natural gas exploration and refining activities account for and estimated 86% of $CO_2$ emissions [3–5] and therefore represent a serious problem for environmental issues related to greenhouse gas emissions and increased global warming, as $CO_2$ is the main cause of anthropogenic global warming. For the NG found in Brazilian pre-salt reserves, these data are even more concerning, as this NG has a higher concentration of $CO_2$, with reserves that have a concentration above 70%; for these cases, the

gas produced is reinjected into the fields as a method of oil recovery [6–8]. Thus, the development of cost-effective $CO_2$-capturing materials is of great importance to the oil industry.

$CO_2$ adsorption using porous solid materials is an alternative to using liquid amines, a process used today in the industry, with zeolites [9], coordinating polymers, activated carbon, and mesoporous silicas [10] the most commonly used substances for this purpose. The first study on silica adsorption was reported in 1995 [11], concluding that silica-based adsorbents have potential for $CO_2$ adsorption. After that study, a wide variety of solids, especially mesoporous silicas, were used for the capture and separation of $CO_2$ [12–17] and synthesized by the hydrothermal method [18,19]. For adsorption tests, the gravimetric method has been widely used in the literature and consists of measuring the adsorbent mass as the adsorption phenomenon occurs [20–24].

Therefore, this research has aimed to synthesize a $CO_2$ adsorbent material from bimodal silica via the hydrothermal method to obtain an adsorbent silica with an ordered structure of the meso–macroporous type. After synthesis, this material was characterized to present the textural properties of the bimodal silica. Then, the adsorbent was tested for its adsorption capacity of $CO_2$, $CH_4$, and mixtures (12%, 25%, 35%, and 50% in Vol) of $CO_2$ in $CH_4$ by the gravimetric method, revealing $CO_2$ selectivity and adsorption capacity.

## 2. Materials and Methods

### 2.1. Synthesis of the Adsorbent

Bimodal silica was synthesized by the hydrothermal method based on the procedure reported in [23]. The materials used were Sodium Silicate, Decan, Hydrochloric Acid, and distilled water as a solvent.

Firstly, 20 mL of water and 3 g of sodium silicate were mixed and stirred continuously at 50 °C to facilitate interactions with the surfactant. A total of 10 mL of decane was then added dropwise to the solution under constant stirring until the desired dispersed phase was obtained. Having formed the emulsion, 5 mL of concentrated hydrochloric acid was added to initiate the silicate hydrolysis reaction. The resulting mixture was stirred for half an hour and introduced into a coated Teflon vessel for hydrothermal treatment at 100 °C for 24 h to allow the silicate to polymerize in the aqueous solution in the presence of the emulsion phase, thereby producing an ordered mesoporous structure containing macropores.

At the end of the hydrothermal treatment, the material obtained was vacuum filtered and then taken to the oven for drying at 120 °C. After drying, the material was washed with a mixture of ethanol and 1 mol·L$^{-1}$ hydrochloric acid solution (1:1) followed by new hydrochloric acid filtration to remove the $Na^+$ ion residues from the sodium silicate. This material was calcined at 550 °C for 5 h to remove the residual surfactant to yield a fine white powdery material. The calcination process was performed in an EDGCON 3P muffle furnace, using a programmed temperature of 10 °C·min$^{-1}$ under a dynamic nitrogen atmosphere of 60 mL·min$^{-1}$ until reaching 550 °C and was kept at this temperature for 4 h. Then, the gas was replaced by an oxidizing atmosphere and held for an additional 1 h.

### 2.2. Characterization

After synthesis, the adsorbent was characterized by X-ray diffraction (XRD), scanning electron microscopy (SEM), specific surface area (BET), and infrared absorption spectroscopy for Fourier Transform (FT-IR) in order to determine its textural properties.

XRD analyses were performed by the powder method on a Bruker D2Phaser apparatus in the 2-theta range: 1–10 (Low) and 5–70 (High).

The specific surface area analyses were performed using Micromeritics ASAP 2020 equipment via the BET method, which consists of the adsorption of $N_2$ at a temperature of 77 K from the adsorption and desorption isotherms.

Scanning electron microscopy (SEM) images were obtained with magnification to allow the visualization of materials at the nanometer scale, and the sample was previously metallized, aiming to make it conductive. The micrographs were obtained using Tescan equipment, model Vega 3.

Fourier Transform Infrared (FT-IR) analysis was performed using the IRAffinity$^{-1}$ model manufactured by Shimadzu in a range from 4000 to 400 cm$^{-1}$.

## 2.3. Obtaining the Adsorption Isotherms

The determination of the adsorption capacity of the adsorbent materials by the gravimetric method involved measuring the adsorbent mass as the adsorbate adsorption phenomenon occurred and was performed in four steps: measurement of the effect of no sample thrust, sample reactivation, measurement of the thrust effect with the sample, and measurement of the adsorbed mass.

For the proposed study, the static adsorption process of the pure gases $CO_2$ and $CH_4$ and their mixtures (12%, 25%, 36%, and 50% $CO_2$ in $CH_4$) was carried out on a Rubotherm magnetic suspension scale, installed at the SENAI Institute of Innovation in Renewable Energy (ISI-ER).

The experimental values obtained were compared and adjusted to the models proposed in the literature (Langmuir and Freundlich) [24].

The Langmuir model considers that the adsorption sites are energetically equivalent, that each molecule of the adsorbate takes one site only, and that they do not interact with each other. This approach can be modelled as follows:

$$q_e = \frac{q_m \cdot b \cdot C_p}{1 + b \cdot C_p} \tag{1}$$

where $q_e$ is the concentration of the adsorbate in equilibrium with the fluid phase (mol/g), $q_m$ is the maximum adsorption capacity (mol/g), $b$ is the equilibrium constant (L/mol), and $C_P$ is the equilibrium concentration of the adsorbate (g/L), which can be determined by the ideal gas law:

$$Cp = \frac{P \cdot MM}{RT} \tag{2}$$

where $P$ is the system pressure (Pa), $MM$ is the adsorbate molecular mass (g/mol), $T$ is the adsorption temperature ($K$), and $R$ is the ideal gas constant (8.314 L·kPa/(mol·K)).

The Freundlich model describes equilibrium on heterogeneous surfaces and allows a logarithmic distribution of the active sites, thereby offering a useful treatment when there is no interaction between the adsorbate molecules; this model can be expressed as follows:

$$q_e = K_e \cdot C_p{}^{1/n} \tag{3}$$

where $K_e$ is the adsorption capacity of the solid (L/g), $1/n$ is a measure of the adsorption intensity, and $qe$ is the concentration of the adsorbate in the solid phase in equilibrium with the fluid phase. Both $Ke$ and $n$ were determined experimentally.

## 3. Results and Discussions

### 3.1. Characterization of the Adsorbents Obtained

Figure 1 presents X-ray diffractograms of the obtained silica at a low angle (1a; 1 to 10°) and at a high angle (1b; 15 to 60°).

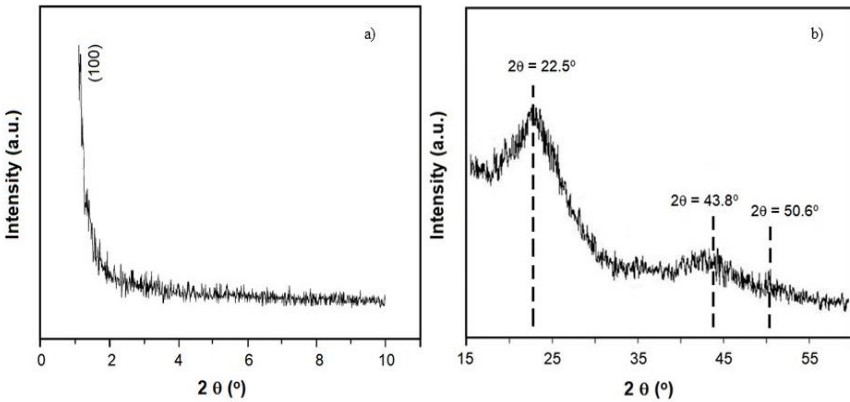

**Figure 1.** X-ray diffractograms of the bimodal silica at a low angle (**a**) and a high angle (**b**).

The results revealed the presence of an acute peak at a low angle and other peaks at a high angle, indicating that silica occurs in a bimodal form. This material can be considered a type of active silica, as it exists in a more reactive phase than the crystalline phase of silica, which do not have peaks that show the long-range order characteristics of the crystal.

In Figure 1a, a peak was observed at about $2\theta = 1.2°$, which was attributed to the plane (100). The shape of the peak shows that an ordered phase of silica was obtained. In Figure 1b, a high intensity Bragg angle centered at $2\theta = 22.5°$ was also recorded. This peak is typical of amorphous silica; however, the presence of two other peaks at $2\theta = 43.8°$ and $2\theta = 50.6°$ indicates the presence of crystallographic phases attributed to alpha-cristobalite and quartz, which are obtained after calcination of the silica (amorphous silica) [25].

Figure 2a,b shows the particle aggregates and particle distribution, respectively. Most $SiO_2$ particles are observed in the 50 to 200 nm size range, thereby characterizing the presence of nanoscale porous materials. These particles have a tendency to form mesoporous aggregates. Macropores are observed between the particle aggregates. For better verification of the porosity of the materials, a computational image processing resource was used; this resource verified the presence of mesoporous and microporous silica, clearly showing the bimodal silica porosity system.

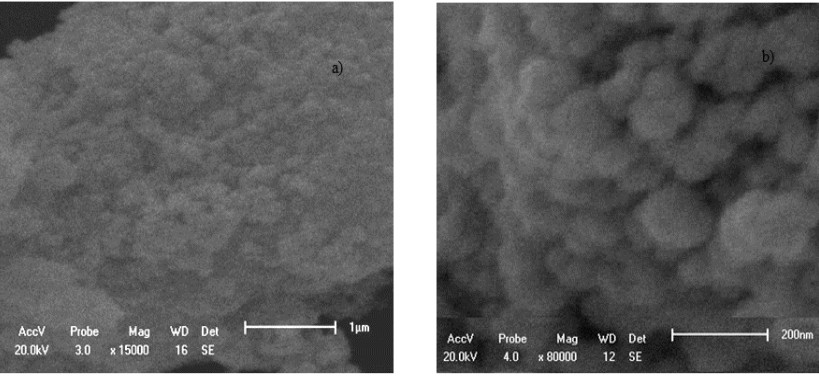

**Figure 2.** SEM image of the bimodal silica showing particle aggregates (**a**) and the particle distribution (**b**).

Figure 3 shows the mesoporous and macroporous ordering in the bimodal silica. Analyzing the image, it appears that the materials obtained feature typically amorphous silica, with an internal structure consisting of structural blocks connected to each other. Despite this amorphous material, this ordering is typically observed due to the nature of the chemical bonding of the silica (-O-Si-O-), with the mesoporous silica forming typically linear cylinder and the macropores originating from interconnected cracks and crevices (Slit Pores). These chemical bonds allow the bimodal silica to react

with other materials around it, facilitating a better adsorption capacity and indicating characteristics beneficial for gas capture and storage.

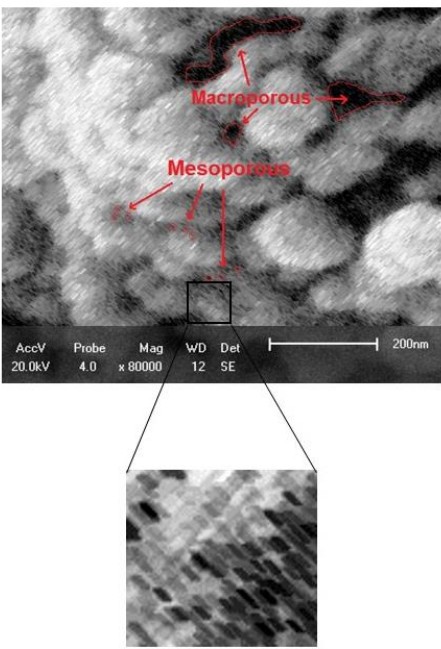

**Figure 3.** Mesopore and macropore ordering in the bimodal silica.

According to the BET equation, Nitrogen adsorption analysis at 77 K obtained the adsorption isotherm, which is shown in Figure 4.

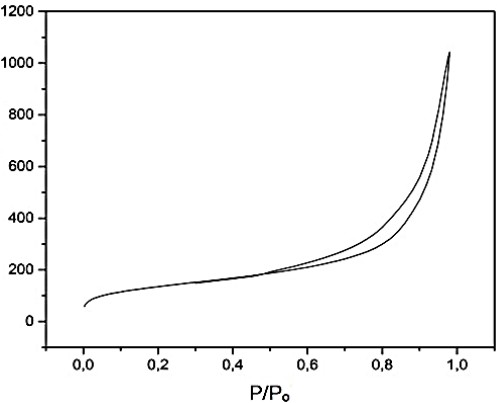

**Figure 4.** Nitrogen Adsorption Isotherm for the Bimodal Silica (obtained at 77 K).

The specific area obtained was equivalent to 230 $m^2/g$, presenting a typical value for ordered silica. The bimodal silica demonstrates a good desorption capacity at low relative pressures ($P/P_0 < 0.3$), which remain almost constant in a relative pressure range from 0.5 to 0.8, with a positive slope in the curve at relative pressures greater than 0.8. This behavior is characteristic of adsorbents that have a uniform pore distribution [26] and presents meso–macroporous pores [15].

During the formation of the $N_2$ adsorption isotherms, two distinct events were observed. The first event shows the amount of gas adsorbed with increasing relative pressure, while the other represents the amount of gas adsorbed in the reverse process. These events are characteristic of solids with large pore sizes (meso and macro), where the adsorbate ($N_2$) evaporation process that occurs inside the pores is different from capillary condensation, with monolayer formation followed by adsorption

from multilayer to isotherm inflection and saturation (which, for bimodal silica, is type I + IV [15,22], according to the IUPAC classification [27]).

By obtaining the results of the nitrogen adsorption and desorption data analysis, it was possible to identify the pore size distribution, as shown in Figure 5a,b, for porosity at 2–5 nm and 5–50 nm, respectively.

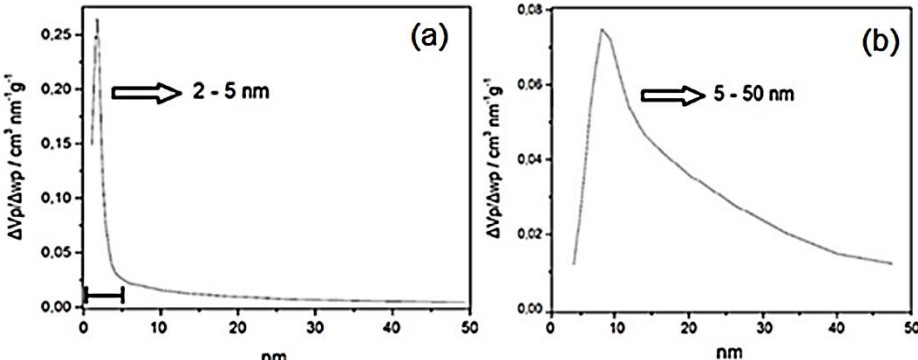

**Figure 5.** Pore volume distribution for bimodal silica in the range of 2–5 nm (**a**) and 5–50 nm (**b**).

In order to verify the chemical bonds in silica, specifically for silane groups, surface water, and the stretches and deformations of the O–Si–O bonds in silica (SiO₂), the infra-red (FT-IR) spectrum of the bimodal silica was analyzed (shown in Figure 6).

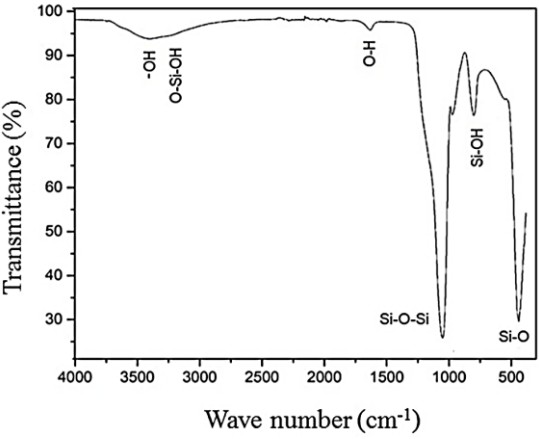

**Figure 6.** Infrared absorption spectrum of the bimodal silica.

The FT-IR spectrum obtained in the range of 4000 to 400 cm$^{-1}$ shows an absorption band at 3435 cm$^{-1}$ caused by the stretching of the -OH bonds in the superficial H₂O molecules. The deformation band of this vibration was observed at 1632 cm$^{-1}$ (H-OH). The shoulder found at about 3245 cm$^{-1}$ can be attributed to the vibrations of the silanol (Si-OH) stretching present in the amorphous silica's structure. The presence of the Si-OH group was proven to be a consequence of water bound to the silica. The strong band observed at 1080 cm$^{-1}$ with a shoulder at about 1190 cm$^{-1}$ is attributed to the asymmetric vibrations of the internal and external tetrahedra of the Si-O-Si bonds. The IR band at 950 cm$^{-1}$ may be assigned to the silanol groups. The gait at about 800 cm$^{-1}$ is attributed to the symmetrical stretching of the Si-O-Si groups, while the gait observed at 475 cm$^{-1}$ is due to O-Si-O strain vibrations.

*3.2. Adsorption Isotherms*

The adsorption capacity of $CO_2/CH_4$ and its mixtures was determined by equilibrium isotherms obtained in the static adsorption process and named according to the pure gases and the $CO_2$ composition in the gas mixtures with $CH_4$, as shown in Table 1.

**Table 1.** Names of the adsorption isotherms.

| GAS | NAME |
|---|---|
| Pure Gas $CO_2$ | $SiCO_2$ |
| Mixture 12% $CO_2$ + $CH_4$ (Balance) | Si12 |
| Mixture 25% $CO_2$ + $CH_4$ (Balance) | Si25 |
| Mixture 36% $CO_2$ + $CH_4$ (Balance) | Si36 |
| Mixture 50% $CO_2$ + $CH_4$ (Balance) | Si50 |
| Pure Gas $CH_4$ | $SiCH_4$ |

In order to study the selectivity of bimodal silica for $CO_2$, the adsorption results of pure $CO_2$ and $CH_4$ gases were evaluated, as shown in Figure 7.

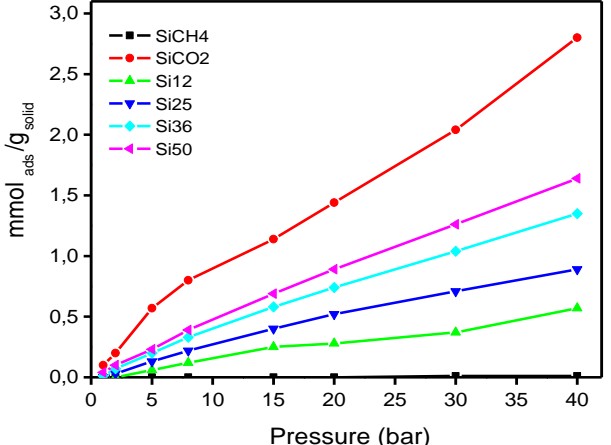

**Figure 7.** Adsorption isotherms.

According to Figure 7, bimodal silica selectivity is notorious for $CO_2$ capture and not for $CH_4$, since it presented practically zero adsorption capacity (0.02 mmol/g) for this gas. This information is extremely important in the development of technologies for the treatment of NG, since the adsorption efficiency for the removal of contaminant $CO_2$ has been reached, making it possible to realize the parameters necessary for commercialization. In order to study the efficiency of bimodal silica as a $CO_2$ adsorbent for pre-salt NG treatment, the compositions of 12%, 25%, 36%, and 50% $CO_2$ in $CH_4$ (balance) were evaluated, as well as the influence of the pressure of these adsorption gases.

For $CO_2$ adsorption in the gaseous composition, the expected result was obtained, with an increase in the mmol amount of $CO_2$ adsorbed by the adsorbent and an increase of this gas in the gas mixture.

Another relevant aspect is related to the influence of gas pressure on adsorption. It is evident that the adsorbed mass increases with an increase in system pressure. These results show the need to perform NG treatment via adsorption at higher pressures, which demonstrates the viability of the sample for the treatment of pre-salt NG.

We also observed a faster adsorption at pressures up to 5 bar, presenting behavior close to the linear model due to the greater availability of the active sites to these pressures. The adsorption results (in mmol) for the $CO_2$/g adsorbent for bimodal silica are shown in Table 2.

**Table 2.** $CO_2$ adsorption capacities obtained at different pressures.

| Presssure (Bar) | Adsorption of $CO_2$ (mmol/g) | | | | |
|---|---|---|---|---|---|
| | $SiCO_2$ | Si12 | Si25 | Si36 | Si50 |
| 1 | 0.10 | 0.00 | 0.00 | 0.01 | 0.04 |
| 2 | 0.20 | 0.00 | 0.03 | 0.07 | 0.10 |
| 5 | 0.57 | 0.06 | 0.13 | 0.20 | 0.23 |
| 8 | 0.80 | 0.12 | 0.22 | 0.33 | 0.39 |
| 15 | 1.14 | 0.25 | 0.40 | 0.58 | 0.69 |
| 20 | 1.44 | 0.28 | 0.52 | 0.74 | 0.89 |
| 30 | 2.04 | 0.37 | 0.71 | 1.04 | 1.26 |
| 40 | 2.80 | 0.57 | 0.89 | 1.35 | 1.64 |
| TOTAL | 9.08 | 1.66 | 2.90 | 4.32 | 5.25 |

Based on the data obtained, the bimodal silica presented very satisfactory results for $CO_2$ adsorption at both low and high pressures due to silica's textural properties, which offer a high surface area containing meso and macropores. In addition, the material has reactivity with other materials around it, facilitating better adsorption capacity.

In order to evaluate the efficiency of bimodal silica in relation to other adsorbents, the adsorption results were compared with those of the literature. To evaluate low pressure adsorbents, Barbosa (2013) [19] and Dantas (2016) [17] developed their studies with commercial solid adsorbents, as shown in Table 3.

**Table 3.** $CO_2$ adsorption capacities at low pressures.

| Adsorbents | Adsorption of $CO_2$ (mmol·g$^{-1}$) | Reference |
|---|---|---|
| $SiCO_2$ | 0.1 | Author |
| MCM-41 | 0.18 | Barbosa (2013) |
| SBA-15 | 0.64 | Barbosa (2013) |
| 10% E-SBA16 | 1.07 | Dantas (2016) |
| 30% E-SBA16 | 0.82 | Dantas (2016) |
| 50% E-SBA16 | 0.77 | Dantas (2016) |

According to the results shown in Table 3, the $CO_2$ adsorption capacity of bimodal silica at low pressures, when compared to the commercially available pure adsorbent materials used in the literature, is slightly lower. For functionalized amines, this difference is much larger because amines favor the adsorption of $CO_2$. However, because bimodal silicas are adsorbent and can be obtained simply with much cheaper materials, they are still very attractive for industry.

When it comes to higher pressures, bimodal silica demonstrated better efficiency compared to the results of Nascimento et al. (2014) [28], as described in Table 4.

**Table 4.** $CO_2$ adsorption capacities obtained at higher pressures.

| Adsorbents | Adsorption Capacity (mmol·g$^{-1}$) | Reference |
|---|---|---|
| $100CO_2$ | 9.10 | Author |
| MCM-48 | 14.89 | Nascimento et al. (2014) |
| SBA-15 | 9.97 | Nascimento et al. (2014) |

For high pressures, the present material was as good as the conventional SBA-15; it was slightly less functional than MCM-48 but still advantageous considering its flexibility, good thermal and chemical stability, and low-cost materials.

After adequate treatment of the data obtained from the microadsorbent, the $CO_2$ adsorption equilibrium isotherms were then adjusted to the Langmuir and Freundlich theoretical models, as shown in Figure 8.

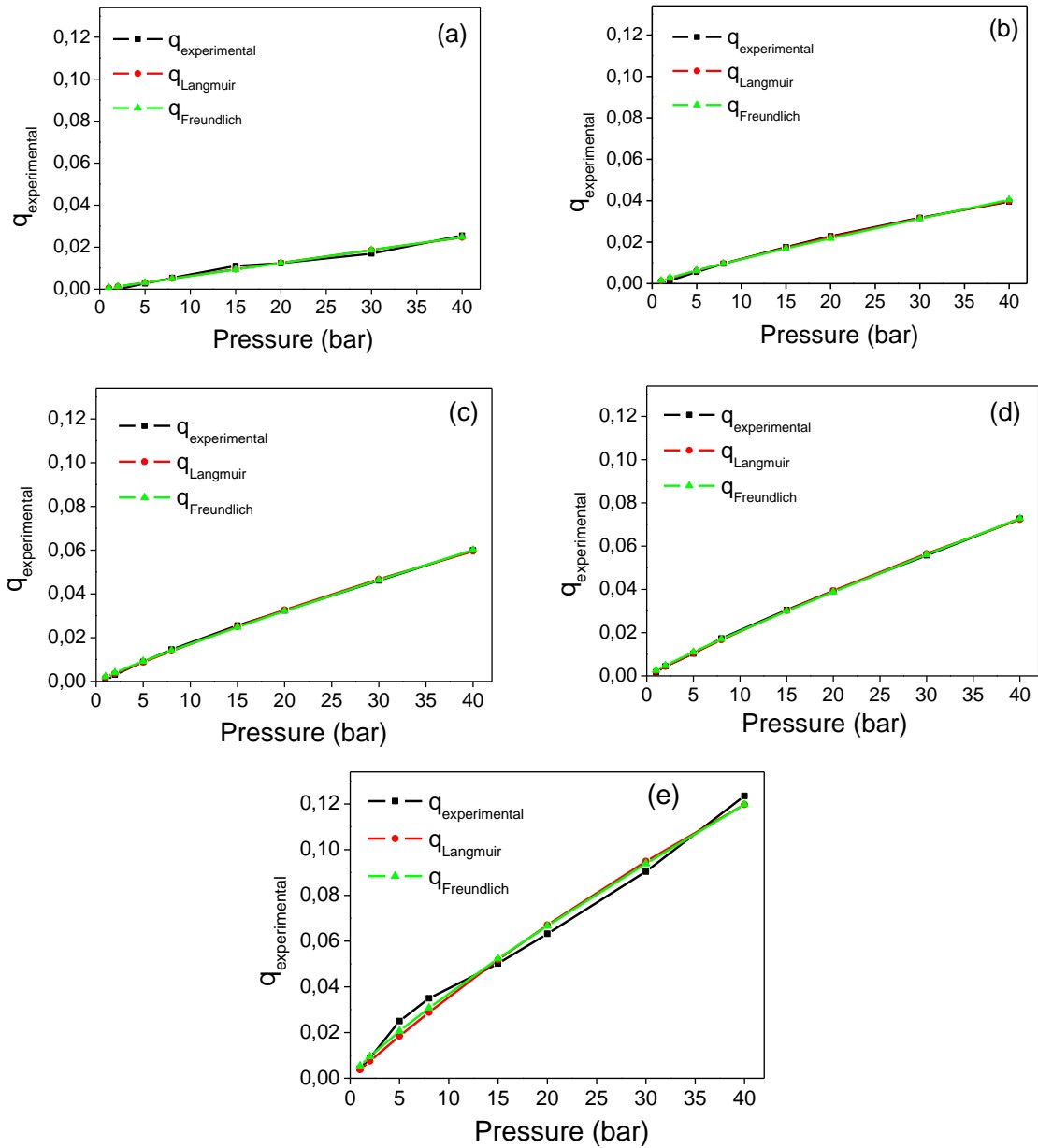

**Figure 8.** Comparison of isotherms of Langmuir and Freundlich Si12 (**a**) Si25 (**b**) Si36 (**c**) Si50 (**d**) and SiCO$_2$ (**e**).

According to the results shown in Figure 8, for low pressure values (0 to 5 bar), all studied isotherms showed linear behavior, which characterizes the beginning of the monolayer formation [15,24]. For values greater than 5 bar, the isotherms presented a profile with a concave curve shape, which was more intense for the 100% CO$_2$ isotherm, thus ensuring that the adsorption of CO$_2$ by the adsorbent is a favorable process.

The estimated parameters in this work for the proposed models are described in Table 5, including the maximum adsorption capacity (q$_m$) and theoretical adsorption capacity in the monolayer (b), based on the Langmuir model; the solid adsorption capacity (K$_F$) and the adsorption intensity parameter (1/n), based on the Freundlich model; and the linear regression coefficients (R$^2$) for the isotherms.

**Table 5.** $CO_2$ adsorption capacities obtained at different pressures.

| Model | Parameters | Si12 | Si25 | Si36 | Si50 | SiCO$_2$ |
|---|---|---|---|---|---|---|
| Langmuir | $q_m$ (g) | 0.626 | 0.177 | 0.334 | 0.4419 | 0.560 |
| | b (L/g) | 0.001 | 0.007 | 0.005 | 0.005 | 0.007 |
| | $R^2$ | 0.98 | 0.99 | 0.99 | 0.99 | 0.98 |
| Freundlich | $K_F$(L/g) | 0.0006 | 0.00152 | 0.00216 | 0.00257 | 0.0053 |
| | 1/n | - | - | - | - | - |
| | $R^2$ | 0.98 | 0.99 | 0.99 | 0.99 | 0.99 |

According to Table 5, both models fit the experimental data under the applied conditions, presenting linear regression coefficients ($R^2$) in the range of 0.98–0.99, indicating that $CO_2$ adsorption was not restricted to the formation of the monolayer and showing that the material presented a surface with heterogeneous adsorption [28]. However, the Langmuir equation was better adjusted to the data obtained experimentally, presenting a qm with values closest to those obtained for the microadsorbent when compared to the Freundlich equation ($K_F$).

We also observed an increase in the $CO_2$ adsorption capacity (higher value of $q_m$) with an increase of its concentration in the gas mixture, thereby demonstrating the viability of using the adsorbent in any concentration of $CO_2$, demonstrating that it can be easily applied to the necessary treatment of pre-salt NG, as proposed in the paper, because it has high concentrations of $CO_2$. In addition to the results obtained, the excellent adsorption capacity of the material was proven compared to the literature results, presenting low pressure adsorption results close to commercial materials of the type MCM-41, which is inferior, according to the results obtained by Barbosa (2013) [19], compared to the SBA16 functionalized with amines [17], since the addition of amines increased the adsorbent potential due to the pure structure lacking active sites to promote interactions with carbon dioxide.

For higher pressures, the material was as good as conventional SBA-15 and slightly less functional than MCM-48 [28]. However, the material is still considered advantageous based on its properties that confer flexibility and good thermal and chemical stability. For the theoretical adjustments, both models fit the experimental data under the applied conditions, showing that the material presented a surface with heterogeneous adsorption [29].

## 4. Conclusions

The development of new hierarchical porosity-based silica-based materials for adsorption processes is a promising direction in nanoscience and nanotechnology. By analyzing the synthesis route used to obtain bimodal silica containing meso and macropores, it has been shown that n-decane hydrocarbon can be successfully used to obtain ordered porosity silica, as well as good particle distribution. The hydrothermal method and synthesis parameters, such as temperature, mixing time, and the amount of the driver, must be pre-adjusted to obtain bimodal silica.

Bimodal silica powder, obtained according to the pre-established parameters, was obtained with a specific surface area of about 230 m$^2$/g and a pore distribution in a mesoporous (2–5 nm) and macroporus (greater than 50 nm) range. The characterization of the material obtained through X-ray diffractometry, scanning electron microscopy, and infrared abortion spectroscopy showed that the amorphous silica material was obtained with ordered porosity in the meso and macropore range, which is considered bimodal, with typical bonds of silica (O-Si-O), silanols (Si-OH), and surface water (H-OH).

Regarding the adsorption of pure $CO_2$, the present material is selective for adsorption only of $CO_2$ and not of $CH_4$, showing that it is an excellent adsorbent for Natural Gas, since $CO_2$ is a contaminant that needs to be removed to meet the requirements and commercial parameters according to current legislation in the country. In the gas mixtures, the expected results were obtained, with an increase in the amount of adsorbed $CO_2$ mmol proportional to the increase of this gas in the gas mixture.

We observed that the influence of gas pressure on adsorption is an important parameter for analysis, as there is a considerable increase in the adsorbed mass when there is an increase in system pressure. These results show the need to perform NG treatment by adsorption using bimodal silica at higher pressures (above 20 bar), which is an advantage when dealing with pre-salt NG. We also observed faster adsorption at pressures up to 5 bar, indicating behavior close to the linear model due to the greater availability of the active sites to these pressures. Additionally, we noticed an increase in $CO_2$ adsorption capacity (higher value of $q_m$) with an increase of its concentration in the gas mixture, demonstrating the viability of using this adsorbent in any concentration of $CO_2$ and indicating that it can be easily applied to pre-salt NG treatment. According to the results above, the proposed adsorbent is an excellent material for the treatment of natural gas and is proven to be effective for the removal of $CO_2$ contaminants.

**Author Contributions:** Conceptualization, A.S.A.; Data curation, F.C.d.C. and P.F.d.N.; Formal analysis, F.C.d.C. and M.R.O.d.S.; Investigation, M.R.O.d.S.; Methodology, F.C.d.C.; Project administration, A.S.A.; Resources, F.C.d.C.; Supervision, A.S.A. All authors have read and agreed to the published version of the manuscript.

**Funding:** This research received no external funding.

**Conflicts of Interest:** The authors declare no conflict of interest.

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
