# Peer review of "The Efficiency of Bimodal Silica as a Carbon Dioxide Adsorbent for Natural Gas Treatment"

_processes, doi:10.3390/pr8030289_

Round 1

Reviewer 1 Report

This paper is interesting but needs major editing before it  can proceed, and the numerous minor language and other errors must be eliminated.

This paper is interesting but needs major editing before it  can proceed, and the numerous minor language and other errors must be eliminated.  I recommend major revisions and resubmission.

Minor

synthesized, promising for use.. English

from 2030  .. do you mean until 2030?

These problems… this is not specific enough  being necessary the effective treatment.. English

Do you mean 86%  of the emissions in Brazil?  Please give a reference.

as CO2 is the main villain of these weather effects… rewrite:.. as CO2 is the main cause of anthropogenic global warming.. or something like that global warming is more than weather effects, e.g. ocean acidification.

Is the 70% the CO2 content, this does seem high, again a reference would be nice.  By the way don’t write Oil industry, just oil industry, it is not a proper noun.  You also periodically do that with names why is Leesa in capitals but not Leal for instance.

By the way why is a company name written in capitals?  Reference4 is not complete.

What is ln 68 and 69, is this a position for a figure?  Is image 1 the same as Figure 1?

The launch into the results and discussion section is not smooth or obvious, the reader will not follow this well, why do we care about image 1?

Table 1, in English there is no acute accent on the a of gas.

Generally it is better to write conclusions when there are more than one of them.

to meet the requirements. commercial parameters.  What does this mean, and why the period?

Reference 19 subscript your chemical symbols.

How would I find reference 16, can you provide a link?

Reference 17 country please, prior to this I thought the only Natal was in South Africa.

The Canadian journal of chemical engineering… why is the rest in lower case as opposed to say Chemical Engineering Journal

Author Response

Point 1: from 2030  .. do you mean until 2030?

 Response 1: Until 2030

Point 2: Do you mean 86%  of the emissions in Brazil?  Please give a reference.

Response 2: Relatório annual Petrobras 2018. Disponível em https://www.investidorpetrobras.com.br/ptb/206/Relatorio_anual_27_03.pdf. Acesso em 08 de novembro de 2019.

BRASIL.MCTI. Emissões Fugitivas de Gases de Efeito Estufa na Indústria de Petróleo e Gás Natural. Relatórios de Referência: Setor Energia. 2º Inventário Brasileiro de Emissões e Remoções Antrópicas de Gases de Efeito Estufa. Brasília, DF: MCTI, 2010.

Machado, Eduardo. (2015). ECONOMIA DE BAIXO CARBONO: AVALIAÇÃO DEIMPACTOS DE RESTRIÇÕES E PERSPECTIVAS TECNOLÓGICAS: PETRÓLEO E PETROQUÍMICA. Disponível em: https://www.researchgate.net/publication/303693117_ECONOMIA_DE_BAIXO_CARBONO_AVALIACAO_DE_IMPACTOS_DE_RESTRICOES_E_PERSPECTIVAS_TECNOLOGICAS_PETROLEO_E_PETROQUIMICA. Acesso em 20/10/2019.

Point 3: as CO2 is the main villain of these weather effects… rewrite:.. as CO2 is the main cause of anthropogenic global warming.. or something like that global warming is more than weather effects, e.g. ocean acidification.

 Response 3: as CO2 is the main cause of anthropogenic global warming

 Point 4: Is the 70% the CO2 content, this does seem high, again a reference would be nice.  By the way don’t write Oil industry, just oil industry, it is not a proper noun.  You also periodically do that with names why is Leesa in capitals but not Leal for instance.

 Response 4: oil industry ok. The dissertations and theses, according to the magazine's citation model, are written with the author's name in capital letters.

Point 5: By the way why is a company name written in capitals?  Reference 4 is not complete.

Response 5: Because SENAI is an acronym, so it is all capitalized. The name of the company is National Service for Industrial Learning. Reference 4 adjusted.

Point 6: What is ln 68 and 69, is this a position for a figure?  Is image 1 the same as Figure 1?

Response 6: Yes, they are the same

 Point 7: The launch into the results and discussion section is not smooth or obvious, the reader will not follow this well, why do we care about image 1?

 Response 7: Adjusted

 Point 8: Table 1, in English there is no acute accent on the a of gas.

 Response 8: Adjusted

Point 9: Generally it is better to write conclusions when there are more than one of them.

Response 9: As there was the synthesis of the adsorbent and it is necessary to characterize it in order to know if we were able to obtain the material with the necessary properties, we have a conclusion for that. Then we carry out the adsorption analyzes that will bring us another result. Thus, we think it is important to insert the conclusions for these two stages. If necessary, we can take it out.

Point 10: to meet the requirements. commercial parameters. What does this mean and why the period?

Response 10: Parameters defined in the legislation in force in Brazil for the sale of natural gas.

Point 11: Reference 19 subscript your chemical symbols.

Response 11:Adjusted

Point 12: How would I find reference 16, can you provide a link?

Response 12:  http://monografias.ufrn.br/handle/123456789/2043

Point 13: Reference 17 country please, prior to this I thought the only Natal was in South Africa.

Response 13: Adjusted

Point 14: The Canadian journal of chemical engineering… why is the rest in lower case as opposed to say Chemical Engineering Journal

 Response 14: Adjusted

Reviewer 2 Report

Dear Authors,

thank You for this interesting paper. In my opinion the topic is interesting with strong dedication to indistrial applications. I found it well-written and well presented and can be therefore published with suggested changes/improvements. Here they are:

1) Apparently there a mistake in Figure 5a - the x-axis ranges 0-50nm however the caption suggests 0-5nm

2) I think Figures 7a and 7b can be merged and enlarged as they present the same data, dont they?

3)Mispell in Tables 3 and 4: AdsorVents

4)Figure 8 is illlegible - please enlarge

5) I suggest to provide the base equatino for Langmuir and Freundlich model referring to Table 5.

6) I would suggest to put the section "Materials and methods" before the results. What should be also added, either to "Materials and methods" or Results, are error analysis (at least included to the graphical results)

7) p.12 l.313 some spelling problem

Best regards

Author Response

Point 1: Apparently there a mistake in Figure 5a - the x-axis ranges 0-50nm however the caption suggests 0-5nm

Response 1: The figure is in the correct scale, from 0-5 nm to a range where it was found as dimensions with these pore sizes

Point 2: I think Figures 7a and 7b can be merged and enlarged as they present the same data, dont they?

Response 2: Adjusted

Point 3: Mispell in Tables 3 and 4: AdsorVents

Response 3:Adjusted

Point 4: Figure 8 is illlegible - please enlarge

 Response 4:Adjusted

Point 5: I suggest to provide the base equatino for Langmuir and Freundlich model referring to Table 5.

Response 5:Adjusted

Point 6: I would suggest to put the section "Materials and methods" before the results. What should be also added, either to "Materials and methods" or Results, are error analysis (at least included to the graphical results)

Response 6: Adjusted

Point 7: p.12 l.313 some spelling problem

Response 7: Adjusted

Reviewer 3 Report

This manuscript describes the usage of bimodal silica as the adsorbent for carbon dioxide. The work is very chaotically written, there is many literal and grammar errors. In several tables many italian words can be seen. Generally, in my opinion, English language requires intensive correction.

For example, see the phrase in the Introduction:

" For the NG found in the Brazilian pre-salt reserves, this data is even more worrying because it has a higher concentration of CO2, with reserves that hsve a concentration above 70%, which for these cases the gas produced is reinjected in the fields, as a method of oil revcover."

other example:

"by the method hydrothermal..."

line 86:

""For better verification of the porosity of the materials, a computational image processing resource was performed through its formatting, where it was verified the presence of mesoporous and macroporous, clearly showing the bimodal silica porosity system."

There is no units on y axi in Fig. 4.

Line 118:

"These events are characteristic of solids with large pore sizes, where the adsorbate evaporation process that occurs inside the pores in different from capillary condensation, with monolayer formation followed by adsorption. from multilayer to isotherm inflection and satuiration, which is bimodal silica is type I +IV..."

Table 1: "misture".

Table 3 and 4: References should be in the brackets (e.g. [1]), listed at the end of the manuscript.

Fig. 8: The graphs are hardly visible.

The mathematical dscription of adsorption process should be described also using non-linear fitting analysis and using other models (Jovanovich, Dubinin-Radushevich). Although Langmuir and Freundlich models are very useful, they were known 100 years ago.

In my opionion, the manuscript requires major revision.

Author Response

Point 1: This manuscript describes the usage of bimodal silica as the adsorbent for carbon dioxide. The work is very chaotically written, there is many literal and grammar errors. In several tables many italian words can be seen. Generally, in my opinion, English language requires intensive correction.

For example, see the phrase in the Introduction:

" For the NG found in the Brazilian pre-salt reserves, this data is even more worrying because it has a higher concentration of CO2, with reserves that hsve a concentration above 70%, which for these cases the gas produced is reinjected in the fields, as a method of oil recover."

other example:"by the method hydrothermal..."

line 86:""For better verification of the porosity of the materials, a computational image processing resource was performed through its formatting, where it was verified the presence of mesoporous and macroporous, clearly showing the bimodal silica porosity system."

Response 1: Adjusted

Point 2: There is no units on y axi in Fig. 4.

Response 2: Adjusted

Point 3: Line 118:

"These events are characteristic of solids with large pore sizes, where the adsorbate evaporation process that occurs inside the pores in different from capillary condensation, with monolayer formation followed by adsorption. from multilayer to isotherm inflection and satuiration, which is bimodal silica is type I +IV..."

Response 3: Adjusted

Point 4: Table 1: "misture".

Response 4: Adjusted

Point 5: Table 3 and 4: References should be in the brackets (e.g. [1]), listed at the end of the manuscript.

Response 5: Adjusted

Point 6: Fig. 8: The graphs are hardly visible.

Response 6: Adjusted

Point 7: The mathematical dscription of adsorption process should be described also using non-linear fitting analysis and using other models (Jovanovich, Dubinin-Radushevich). Although Langmuir and Freundlich models are very useful, they were known 100 years ago.

Response 7: According to the literature review carried out by the group, for the type of adsorbent synthesized, these are the models that best fit. However, for the next tests the suggested models will be used. We appreciate the guidance.

Round 2

Reviewer 1 Report

Please do not define standard chemical symbols which are part of the standard scientific nomenclature and which have existed before any of us were born.  This is an immensely annoying and unfortunate practise and if you insist on it then my recommendation is to reject your paper.  Otherwise I can recommend acceptance with only minor revisions.

Do not capitalize natural gas, there is no need.  By contrast do capatilize Brazilian everywhere in the manuscript.

What is VETEC HCl?

What is bimodal silica?

It is unreasonable to write R2 values to four or five decimal places.

Line 420 subscript your chemical symbols.

Conclusions are better than conclusion as there are more than one of them.

What is an amount of driver.

Do not write Bar, actually you should use SI but if you want to use bar then write it in lower case.

Reference 25 lower case for SiO2

Author Response

Point1: Please do not define standard chemical symbols which are part of the standard scientific nomenclature and which have existed before any of us were born.  This is an immensely annoying and unfortunate practise and if you insist on it then my recommendation is to reject your paper.  Otherwise I can recommend acceptance with only minor revisions.

Response 1: Ok, sorry

Point 2:Do not capitalize natural gas, there is no need.  By contrast do capatilize Brazilian everywhere in the manuscript.

Response 2: Ok

Point 3: What is VETEC HCl?

Response 3: Hydrochloric acid from the vetec brand, already adjusted in the text

Point 4: What is bimodal silica?

Response 4: It is an adsorbent for CO2 based on silca, bimodal silica, containing meso and macropores, synthesized in this research

Point 5: It is unreasonable to write R2 values to four or five decimal places.

Response 5: Adjusted

Point 6: Line 420 subscript your chemical symbols.

Response 6: Adjusted

Point 7:Conclusions are better than conclusion as there are more than one of them.

Response 7: Adjusted

Point 8:What is an amount of driver.

Response 8: Is the amount of decane used in the synthesis

Point 9: Do not write Bar, actually you should use SI but if you want to use bar then write it in lower case.

Response 9: Adjusted

Point 10: Reference 25 lower case for SiO2

Response 10: Adjusted

Reviewer 3 Report

The corrected manuscript can be accepted for publication.

Author Response

Point 1: I would not like to sign my review report 

Response1: Ok, thank you very much